­Blue mussel (Mytilus spp.) cultivation in mesohaline eutrophied inner coastal waters: mitigation potential, threats and cost effectiveness

Ritzenhofen Lukas lukas.ritzenhofen@io-warnemuende.de 1 2
Buer Anna-Lucia 1
Gyraite Greta 1 2
Dahlke Sven 3
Klemmstein Annemarie 1
Schernewski Gerald 1 2
1 Leibniz-Institute for Baltic Sea Research, Warnemünde , Rostock , Germany
2 Marine Research Institute, Klaipeda University , Klaipeda , Lithuania
3 Biological Station Hiddensee, University of Greifswald , Greifswald , Germany
O’Connor Wayne
Electronic publication date: 2021 May 20
Publication date: 2021
Volume: 9
Electronic Location ID: e11247
Received 2020 Oct 28; Accepted 2021 Mar 18
Copyright: ©2021 Ritzenhofen et al.
Copyright year: 2021
Copyright holder: Ritzenhofen et al.
License: This is an open access article distributed under the terms of the Creative Commons Attribution License, which permits unrestricted use, distribution, reproduction and adaptation in any medium and for any purpose provided that it is properly attributed. For attribution, the original author(s), title, publication source (PeerJ) and either DOI or URL of the article must be cited.
License URL: https://creativecommons.org/licenses/by/4.0/

Keywords: Mussel cultivation, Eutrophication, Mitigation, Vibrios, Cost-Effectiveness

Funding: BONUS OPTIMUS 03A0020A BONUS Art 185 European Union’s Seventh Program for research, technological development and demonstration Baltic Sea national funding institutions Open Access Fund of the Leibniz Association Doctorate Study Programme in Ecology and Environmental Sciences, Klaipeda University The work was financially supported by the project BONUS OPTIMUS (03A0020A). The project has received funding from BONUS (Art 185) funded jointly by the European Union’s Seventh Program for research, technological development and demonstration, and from Baltic Sea national funding institutions. The publication of this article was funded by the Open Access Fund of the Leibniz Association. This study was supported by the Doctorate Study Programme in Ecology and Environmental Sciences, Klaipeda University (for Lukas Ritzenhofen and Greta Gyraite) There was no additional external funding received for this study. The funders had no role in study design, data collection and analysis, decision to publish, or preparation of the manuscript.

==============================
The EU-water framework directive (WFD) focuses on nutrient reductions to return coastal waters to the good ecological status. As of today, many coastal waters have reached a steady state of insufficient water quality due to continuous external nutrient inputs and internal loadings. This study focuses first on the current environmental status of mesohaline inner coastal waters to illustrate their needs of internal measures to reach demanded nutrient reductions and secondly, if mussel cultivation can be a suitable strategy to improve water quality. Therefore, nitrogen, phosphorus, chlorophyll a, and Secchi depth of nine mesohaline inner coastal waters in north east Germany were analyzed from 1990 to 2018. Two pilot mussel farms were used to evaluate their effectiveness as a mitigation measure and to estimate potential environmental risks, including the interactions with pathogenic vibrio bacteria. Further, estimated production and mitigation potential were used to assess economic profitability based on the sale of small sized mussels for animal feed and a compensation for nutrient mitigation. The compensation costs were derived from nutrient removal costs of a waste water treatment plant (WWTP). Results show that currently all nine water bodies do not reach the nutrient thresholds demanded by the WFD. However, coastal waters differ in nutrient pollution, indicating that some can reach the desired threshold values if internal measures are applied. The mitigation potential of mussel cultivation depends on the amount of biomass that is cultivated and harvested. However, since mussel growth is closely coupled to the salinity level, mussel cultivation in low saline environments leads to lower biomass production and inevitably to larger cultivation areas. If 50% of the case study area Greifswald Bay was covered with mussel farms the resulting nitrogen reduction would increase Secchi depth by 7.8 cm. However, high chlorophyll a values can hamper clearance rates (<20 mg m−3 = 0.43 l h−1 dry weight g−1) and therefore the mitigation potential. Also, the risk of mussel stock loss due to high summer water temperatures might affect the mitigation potential. The pilot farms had no significant effect on the total organic content of sediments beneath. However, increased values of Vibrio spp. in bio deposits within the pilot farm (1.43 106 ± 1.10 106CFU 100 ml−1 (reference site: 1.04 106 ± 1.45 106 CFU 100 ml−1) were measured with sediment traps. Hence, mussel farms might act as a sink for Vibrio spp. in systems with already high vibrio concentrations. However, more research is required to investigate the risks of Vibrio occurrence coupled to mussel farming. The economic model showed that mussel cultivation in environments below 12 PSU cannot be economic at current market prices for small size mussels and compensations based on nutrient removal cost of WWTPs.

Introduction

Globally, water quality of coastal ecosystems has deteriorated because of anthropogenic eutrophication. The European Water Framework Directive (WFD, 2000/60/EC) and the European Marine Strategy Framework Directive (MSFD, 2008/56/EC) were implemented to reduce eutrophication, and the Baltic Sea Action Plan (BSAP) expanded and implemented the WFD in the Baltic Sea (Schernewski et al., 2015). The overall aim is to restore the good ecological status (GES) of aquatic systems by 2027 (WFD, 2000/60/EC). Indicators of the ecological status are submersed macrophytes, zoobenthos, and phytoplankton; nutrient concentrations and water transparency serve as supporting parameters (WFD, 2000/60/EC).

In the Baltic Sea, approximately 86% of coastal waters are below a good trophic status (HELCOM, 2016). In Germany, the water quality of shallow, enclosed, and semi enclosed water bodies are currently rated unsatisfactory or poor (UBA, 2017). However, semi enclosed systems could still reach the turning point and return to a mesotrophic state, if nutrient loads and internal nutrient concentrations are reduced substantially (Berthold et al., 2018; Friedland et al., 2019a).

Several internal mitigation measures are already under discussion for the Baltic Sea, including geoengineering approaches like large scale oxygen ventilation (Stigebrandt et al., 2014) and P-precipitation (Rydin et al., 2017), as well as biomass harvesting in terms of commercial and non-commercial fish, bivalves, or water plants (Lindahl, 2012; Petersen et al., 2014; Karstens et al., 2018). Nevertheless, many internal measures are controversially discussed. While some have been associated with high costs and lack feasibility, others are accompanied with undesirable side effects to the aquatic environment, such as damage to benthic habitats or substantial changes in food webs (Naylor, Williams & Strong, 2001; Stadmark & Conley, 2011; Wikström et al., 2020).

According to many authors, mussel cultivation seems to be one of the most promising measures (Lindahl, 2012; Petersen et al., 2014; Holbach et al., 2020; Kotta et al., 2020a; Kotta et al., 2020b). Besides removing nutrients through biomass harvest, mussels can benefit the water transparency as a result of their filter feeding activity (Nielsen et al., 2016, Schröder et al., 2014) and can provide an economically interesting product. Longlines are the common practice of mussel cultivation in the Baltic (Taylor et al., 2019) and, compared to bottom culture, they are most promising for nutrient removal (Buer et al., 2020a; Buer et al., 2020b; Wiles et al., 2006; Filgueira, Grant & Petersen, 2018).

Mussel mitigation has its limitations though, due mainly to the physiology of mussels. Within the Baltic Sea, only Mytilus spp. and Dreissena spp. are suitable for longline cultivation, as they use byssus threads to attach to the provided collector material (Stybel, Fenske & Schernewski, 2009; Friedland et al., 2019b).

While Mytilus spp. is distributed from full marine conditions to a salinity minimum of 4.5 PSU (Darr, Gogina & Zettler, 2014; Schiele, Darr & Zettler, 2014; Schiele et al., 2015; Larsson et al., 2017), Dreissena spp. occurs in Baltic lagoons and bays with up to 5.0 PSU (Ackerman et al., 1994; Fenske, 2005; Stybel, Fenske & Schernewski, 2009).

Even though mussel cultivation has a relatively small negative environmental impact when compared with coastal fin fish aquaculture (Newell, 2004; Dumbauld, Ruesink & Rumrill, 2009), it is still associated with negative effects on the environment when one looks at intensive production units and the resulting bio deposition (Weise et al., 2009). Increased bio deposition from suspended mussel cultivation can result in organic enrichment of the sediments below mussel farms, increasing benthic mineralization rates and sediment oxygen demands, as well as the release of sediment bound nutrients (Giles, Pilditch & Bell, 2006; Richard et al., 2007). These environmental pressures are expected to be stronger in shallow eutrophic coastal waters due to the limited dispersal of bio active material and preloaded sediments.

Another threat posed by mussels is the spread of pathogens (Potasman, Paz & Odeh, 2002).

Most cultivation sides and consumption-mussels are monitored for fecal indicator bacteria (Escherichia coli & Enterococci), but neglect the risk of other associated bacteria, such as Vibrio (Rincé et al., 2018). The increasing appearance of the pathogenic Vibrio spp. bacteria in eutrophic coastal waters has led to a strong public reaction (Ruppert et al., 2004; Gyraite, Katarzyte & Schernewski, 2019; Metelmann et al., 2020). Moreover, previous studies show that filter feeders can accumulate bacterial pathogens and act as a reservoir of Vibrio bacteria (Cavallo & Stabili, 2002; Stabili, Acquaviva & Cavallo, 2005); thus, mussel cultivation in eutrophic environments may promote Vibrio abundancy and increase the risk for humans and other marine species (Callol et al., 2015; Ina-Salwany et al., 2019).

This study investigates the ecological status of inner coastal waters and their demands to reduce nutrients to reach the target values required to ensure the GES, and if blue mussel cultivation can be a suitable mitigation tool for this purpose.

The objectives of this study are to (a) document the long term development of nutrients, chlorophyll a concentrations, and water transparencies in mesohaline inner coastal waters of the south western Baltic, to identify quantitative needs of nutrient and phytoplankton removal; (b) present the ecological results of a test mussel farm in Greifswald Bay; (c) calculate and assesses costs associated with mussel cultivation as a mitigation measure; and (d) discuss the quantitative potential and cost-effectiveness of blue mussel cultivation as a means to reach GES within the WFD and the potential of farming in other mesohaline inner coastal waters.

Methods & Study Site

Study site

The coast of the federal state Mecklenburg-Vorpommern is characterized by several bays, backwaters (Haffe), estuaries and lagoons embedded by islands and peninsulas that form the inner coastal waters (Fig. 1). Salinity classifies two types of inner coastal waters for the German Baltic coast: oligohaline (0.5 to 5–6 PSU) and mesohaline inner coastal waters (5–6 to 18–20 PSU) (Schernewski et al., 2004).

Wismar Bay and Salzhaff in the west are both part of the Bay of Mecklenburg with an intense water exchange with the open Baltic Sea (Schlungbaum & Baudler, 2001).

The Warnow Estuary, surrounded by the city of Rostock, is the transition zone between the river Warnow and the Baltic Sea. Salinity levels in the Warnow Estuary are strongly influenced by stratification caused by river water and Baltic water inflow.

The Barther Bodden is part of the Darß-Zingst Bodden chain. Its salinity level is indirectly influenced by water exchange from the open Baltic Sea and the oligohaline Bodstedter and Saaler Bodden (Schumann et al., 2006). The Strelasund is a strait that separates the Island of Rügen from the German mainland. The Westrügensche Bodden, Greifswalder Bodden (GWB), Kleiner Jasmunder Bodden, and Nordrügensche Bodden all form the coastal lagoons around the Island of Rügen. Salinity levels of coastal lagoons vary according to direct or indirect water exchange with the Baltic Sea (Fig. 1, Table 1).

Figure 1 Map of the study site.

Mesohaline inner coastal water bodies (blue areas, labeled in red) including monitoring stations of the State Agency for Environment, Nature Conservation and Geology Mecklenburg-Vorpommern (LUNG) (red dots) providing environmental data for every water body in the study area (where multiple measuring stations occur, data of these stations are pooled (marked by a dotted line)). Green dots represent the two pilot mussel farms.

The GWB is the biggest inshore basin located in north east Germany, limited by the mainland to the south and the Island of Rügen to the West. The average water depth is 5.8 m, with a maximum depth of 13.6 m. Sediment types consist mostly of mud, sand, and clay gravel-mixture. Based on the lack of hard substrate (Kanstinger et al., 2018), only a limiting and or fluctuating) Mytilus spp. stock can be expected. It is most likely that mussel larvae enter GWB from mussel beds outside, which can be found on reef structures around Rügen Island (Darr, Gogina & Zettler, 2014). Salinity levels range from 6–8 PSU. Water temperatures are highly influenced by seasonal atmospheric temperature regimes.

The GWB plays an important role, both ecologically and economically. While tourism and coastal fishing are essential for local employment (Obenaus & Köhn, 2002; http://www.statistik-mv.de), the GWB serves also as an important spawning ground of the herring stock in the western Baltic Sea (Kanstinger et al., 2018). High nutrient inputs in the past have led to heavy eutrophication (Munkes, 2005). As a result, submersed macrophyte cover declined from 90% to 15% and the GWB turned into a phytoplankton-dominated ecosystem (Munkes, 2005). Altogether, it seems that present mitigation measures in the catchment area are not sufficient to return to a GES and supporting internal measures are needed.

Table 1 Main characteristics of the investigated coastal water bodies.

Data for surface area, water depth, salinity and water exchange are extracted from Schiewer (2007) and Schlungbaum & Baudler (2001).

Water body	Area [km2]	Mean depth [m]	Salinity [PSU]	Water exchange rate [a−1]	
Wismar Bay	170	5.5	11	0.01	
Salzhaff	21	2.3	11	0.01	
Unterwarnow	12.5	4	12	0.03	
Barther Bodden	19.4	1.6	7	0.13	
Strelasund	64.4	3.9	10	0.04	
Westrügenscher Bodden	171.3	1.8	10	0.14	
Greifswalder Bodden	514	5.8	7.5	0.1	
Kleiner Jasmunder Bodden	28.4	2.8	[<7]	0.14	
Nordrügenscher Bodden	124	3.5	9	0.14	

Hydro-chemical and biological data analysis

The hydro-chemical and biological data were provided by the State Agency for Environment, Nature Conservation and Geology Mecklenburg-Vorpommern (LUNG). Monitoring stations provide monthly data for total nitrogen (TN) and phosphorus (TP), dissolved inorganic nitrogen and phosphorus, Chl. a concentration, and Secchi-depth for the period from 1990-2018. Dissolved inorganic nitrogen (DIN) is the sum of nitrate, nitrite, and ammonium. Samples for dissolved nutrients were stored dark until filtration through cellulose-acetate filters (pore size 0.45 µm) followed by freezing. Dissolved inorganic phosphorus (DIP) was determined as soluble reactive phosphorus by using the molybdenum blue method (Strickland & Parsons, 1972). The percentage share of non-bioavailable DIP was not determined. TP and TN were digested in a microwave with peroxodisulfate (DIN11905-1 1998). Total and dissolved nutrients were measured in a continuous flow analyser (Skalar Inc., later Alliance Instruments (Malcolm-Lawes & Wong, 1990; DIN13395 1996; DIN11905- 1 1998; DIN11732 2005). Chl. a was determined fluorometrically (665 nm) after filtration (GF/F, 0.7 µm), extraction with ethanol, and acidification (ISO 10260:19922).

To evaluate whether internal mitigation measures are suitable, time series analysis of total nutrients (TN & TP), Chl. a concentration, and Secchi depth were conducted for the period between 1990 and 2018. The data was split into three periods and average annual concentrations from 1990–1999, 2000–2009, and 2010-2018 were compared. The time series were analyzed using the Mann-Kendall test to identify whether the measured values of the time series are randomly distributed or if trends occur in the data. The level of significance was set to 0.05. Afterwards, values were compared with new WFD target concentrations (Schernewski et al., 2015).

A general approach to managing eutrophication is the control of nutrient loads (N, P) essential for primary production. It has to be known weather phytoplankton biomass responds to changes in nitrogen or phosphorous in the water column. To investigate this, monthly DIN/DIP ratios and Chl. a concentration for all water bodies were compared. Simple linear regression was used to describe the monthly relationship between TN or TP to Chl. a concentration. To investigate how nutrient reduction potentially improves water transparency, simple linear regression models were used as well. Data was ln-transformed when necessary.

Pilot blue mussel farms setup

Pilot farm 1 is located in the Hagensche Wiek, a bay in the north eastern part of GWB (Fig. 1). The cultivation unit consisted of five parallel longlines (Fig. 2). Each longline moored in a depth between 5.0 and 6.0 m using dead weight anchors of 250 kg. Every longline was attached with 50 m collector material, consisting of one mm thick and 50 mm wide polypropylene bands reaching up to 3.5 m water depth. Pilot farm 1 was deployed in April 2017 and was operated until October 2019. During the operating period, the collector material was used for spat collection and mussel grow out.

Figure 2 Design of the two pilot longline mussel farms located in GWB and Wieker Bay.

(A) Pilot farm 1 located in GWB. (B) Pilot farm 2 located in Wieker Bay. Credits: Lukas Ritzenhofen, Sven Dahlke.

A smaller test farm (pilot farm 2) was located in the Wieker Bay (WB) (Fig. 1). The setup consists only of two longlines, each 10 m long (Fig. 2). The collector band material matches the one of pilot farm 1 and the loops extended from 1 m to 1,5 m. Pilot farm 2 was deployed in June 2017.

Field and laboratory experiments

Mesocosm experiments were carried out at GWB and WB to estimate the individual clearance rate at high Chl. a concentrations of Mytilus spp. The reduction of Chl. a was measured using an in vivo fluorometer (Algae Torch, bbe-Moldaenke, Germany).

Additional laboratory experiments tested different Chl. a concentrations and their effect on the clearance rate (CR) of Mytilus spp. During the laboratory experiments, different concentrations of Rhodomonas spp. were added to a well-mixed and aerated aquarium with defined volume of artificial seawater and a group of mussels (n = 20). The reduction of Chl. a concentration was measured using a handheld fluorometer (AquaFluor, Turner Designs, USA). The clearance rate was determined by the exponential decrease in Chl. a concentration as a function of time using the equation CI = a*(V/n), where V = volume of water, n = number of filtering mussels, and a = slope of the regression line in a semi-ln plot of the reduction of Chl. a. Measurements were repeated three times. Control runs without mussels were conducted to evaluate possible sedimentation. Mesocosm and laboratory experiments were conducted at water temperatures of 20 °C.

Nutrient harvest yields were estimated by the nitrogen and phosphorus content of cultivated blue mussels. Mussel tissue and shells of all individuals were pooled, freeze- dried, and ground prior to analyzing carbon and nitrogen with an auto-analyzer (Elementaranalysator EA 3000). Phosphorous was determined in mussel ash using an alkaline persulfate oxidation after Hansen & Koroleff (1999).

The bio deposition of faeces and pseudo-faeces was measured with sediment traps. The traps were deployed within the GWB farm and at a reference point 100 m SW (N54°18,839; E13°40,728) in May 2018. Additionally, sediment traps were deployed at a commercial mussel farm (Farm size = 1 ha) in the Kiel Fjord. Trap contents, including all suspended particles, were filtered onto Whatman GF/F filters, dried, and weighed. Total organic content (TOC) was measured as loss on ignition. Furthermore, sediment samples were taken within GWB farm and the reference point. Subsequently, TOC was measured as loss on ignition of dry matter.

To measure the Vibrio spp. abundance within the pilot mussel farm in GWB and the reference location, samples were taken of surface water, sediments, sediment traps and mussels.

The total number of Vibrio spp. from water samples was determined by filtering water volumes of 10 and 50 mL through 0.45 μm-pore-size mixed cellulose ester filters (MontaMil® Membrane Filters, Frisenette ApS, Knebel, DK). All filters were placed onto thiosulphate-citrate-bilesalts-sucrose agar plates (TCBS agar) (SIGMA-ALDRICH, Missouri, US1A) and incubated for 24 h at 37  °C. Sediment and sediment trap samples were serially diluted and plated on TCBS agar. Fresh mussel samples collected at the mussel farm were transported in cooling boxes and processed within 8 h of collection. Sample preparation and homogenization was done according to the CEFAS Standard Operating Procedure—Detection of Vibrio parahaemolyticus in Bivalve Molluscan Shellfish (http://www.crlcefas.org). Serially diluted sample homogenate was plated on TCBS agar. After 24 h of incubation, colonies were counted and the number of total Vibrio determined as colony forming unit (CFU) 100 mL−1 for water, CFU 100 g−1 for sediment samples and CFU g−1 mussel tissue.

Economic calculations

Data for the economic calculations are based on the investment and maintenance costs of pilot farm 1 in GWB. Since pilot farm 1 was a scientific mussel farm and therefore without annual running cost (that would otherwise be faced by a commercial business), supplementary data were derived from a commercial mussel farm in Kiel Fjord (Krost et al., 2011).

Krost et al. (2011) predicted annual yields of 25 t and 100 t of mussels for human consumption after 18 months. Annual running costs include anchoring and foundation, longlines, buoys, floaters, socks, collectors, and staff. The economic lifetime varied between 1 and 5 years. Investment costs covering a boat and machinery (both with an assumed lifetime of 5 years), as well as a land station, were summarized under annual investment costs. Since staff is the dominating cost factor, the salaries from the year 2010 (Krost et al., 2011) were recalculated for the year 2020 assuming an increase of 30% (the statistical average salary increase in Germany). With increasing farm size, we assumed that staff and investment costs decrease steadily per produced ton of mussels; this decrease is relatively high in smaller farms and negligible in very large units.

The salinity in GWB only allows for the production of mussels for mussel meal or feed, harvested after about 12 months of cultivation time. As a consequence, we assumed lower production costs with a reduced staff effort of 50% and 2% lower investment costs. These numbers are based on our own experiences with our two small pilot farms and verified by discussions with experts from the commercial mussel farm sector and literature.

The annual production of feed mussels in GWB is assumed to be 0.278 kg m−3 cultivation volume. This data is based on a model simulation study in Buer et al. (2020) and verified by our experimental farms. Assuming a cultivation volume from the surface to a depth of 3 m, and taking into account the relatively low productivity, a farm producing 100 t mussels per year would cover a surface area of about 120 ha. The nitrogen (phosphorus) content in mussels is assumed to be 0.008 (0.0006) kg kg−1 fresh mussels, based on the cultivated mussels in GWB.

In addition to the production of feed mussels, we assumed a compensation for nutrient extraction based on the removal costs of a wastewater treatment plant (WWTP). The revenue for nitrogen and phosphorus removal is set at 30,000 € t−1 and 80,000 € t−1 (COWI, 2007; Gren, Jonzon & Lindqvist, 2008; Hautakangas et al., 2014).

Results

Bio-chemical status of coastal waters

Although all nine study sites were similar, their ecological status and its temporal development differed strongly (Fig. 3). The majority had average annual TN concentrations between 20–50 µmol l−1, TP concentrations between 1–2 µmol l−1, and Chl. a concentrations between 5–20 mg m−3. Consequently, all coastal waters were regarded as eutrophic, their ecological status was classified as unsatisfactory (LUNG 2013), and a need for measures to improve water quality was apparent. During the 1990s, the technical improvement of WWTPs caused a strong reduction of nutrient loads to coastal waters in the Federal State of Mecklenburg-Vorpommern, Germany. The largest five WWTPs reduced TP loads by more than 95% and TN loads by over 80% (LUNG 2013). As a result, the riverine loads declined by about 70% (TP). A significant decline in TN loads did not take place during the last 30 years. The riverine loads closely correlate with the river discharge. This is true for the TP loads after 1996, as well (LUNG 2013). In seven of the coastal waters, the load reductions during the 1990s caused a decline in TN and TP annual concentrations (Fig. 3), but only two showed reductions in Chl. a concentrations and only three increased water transparency. After 2000, the nutrient concentrations in the coastal waters did not indicate significant changes. Since the riverine loads were largely stable during the past decades, it can’t be expected that a reduction in nutrients in coastal waters will take place in the future without new, additional measures. Further, it seems unrealistic that additional external load reductions will be implemented

Figure 3 Development of total nutrients, Chl. a. concentration and Secchi depth between 1990 and 2018 in the different water bodies of the study site.

Total nitrogen in red, total phosphorus in blue, Chl. a concentration in green and Secchi depth in grey. Deviation between the current status (2020-2018) (spacing and labeled deviation in red) and the target values demanded by the WFD (green lines).

After 2000, the Chl. a concentrations did not show a significant decline. In most coastal waters, positive effects on water transparency were not visible either. Exceptions were the Barther Bodden, the Jasmunder Bodden, and the Warnow Estuary. It is very likely that the status of the two Bodden waters improved as a result of reduced diffuse agricultural nutrient loads. The Warnow Estuary, as a shipping channel, was subject to several dredging activities. This increased the water exchange with the Baltic Sea and reduced the nutrient and Chl. a levels. In general, the nutrient and Chl. a levels seem directly related to the water exchange time and the shallowness of systems. A low water exchange time and a large euphotic zone (low average water depth) are likely reasons for the very high nutrient and Chl. a concentrations in the Kleiner Jasmunder and the Barther Bodden, as well. On the other hand, the relatively strong reductions in nutrients and Chl. a concentrations in both water bodies over the last decades indicate a distinct reaction to changes in loads. The water exchange time and the average depths are parameters that determine the effectiveness of internal measures in coastal waters.

The sensitivity of aquatic systems is one factor that determines how promising a potential internal measure will be. Another factor is the present state of pollution and its deviation from the good ecological status. Figure 3 indicates the desired threshold values, according to the WFD for each coastal water. Wismar Bay and Salzhaff are already close to a good status and efforts to reduce diffuse loads might already be enough to meet the threshold. While for the Warnow Estuary, the West/Nord Rügener Bodden, and Strelasund and Greifswalder Bodden it seems possible to meet the good status, it would require additional internal measures. Thanks to its size and importance for fishing and recreating, the Greifswalder Bodden is the best case study for assessing the suitability of mussel farming to improve ecological status.

Seasonal nutrient and chlorophyll relationships

During the winter, primary production in Baltic coastal waters is limited by temperature and light. The increase of Chl. a concentrations and decrease of dissolved inorganic nutrient concentrations that takes place during the spring and summer indicates that nutrients become a potentially limiting resource for phytoplankton biomass in most water bodies (Table 2). In all coastal waters, DIN concentrations strongly decline between late winter and summer. For DIP, the decline takes place earlier and by March concentrations are low. The molar N/P ratio of 16 (according to Redfield) provides a rough estimate for which nutrient is limiting primary production.

Table 2 Seasonal dissolved inorganic nitrogen (DIN), phosphorus (DIP) and Chl. a. concentration in the different water bodies of the study site.

(A) Spring = March-April. (B) Summer = May-September. (C) Fall = October-November. DIN:DIP ratios indicate limitation of either nitrogen (bold numbers) or phosphorus (italic numbers).

A	Spring	
	March	April	
	DIN [µmol/l]	DIP [µmol/l]	DIN/DIP	Chl.a [mg m−3]	DIN [µmol/l]	DIP [µmol/l]	DIN/DIP	Chl.a [mg m−3]	
Wismar Bay	9.94	0.36	27	8.39	2.28	0.15	16	1.49	
Salzhaff	73.85	0.23	325	7.36	34.63	0.09	391	2.42	
Warnow Estuary	117.36	0.60	196	6.63	40.70	0.35	116	7.16	
Barther Bodden	92.87	0.13	720	36.19	26.44	0.12	212	77.68	
West Ruegenscher Bodden	26.76	0.40	68	7.44	4.16	0.16	26	8.38	
Strelasund	16.60	0.10	159	19.36	8.85	0.09	93	12.03	
Greifswalder Bodden	26.70	0.11	249	39.59	10.24	0.10	103	18.97	
Kleiner Jasmunder Bodden	40.88	0.20	209	76.49	9.34	0.26	36	111.44	
Nord Ruegenscher Bodden	13.70	0.14	101	15.30	5.55	0.08	67	19.60	
B	Summer	
	May	June	July	August	September	
	DIN [µmol/l]	DIP [µmol/l]	DIN/DIP	Chl.a [mg m−3]	DIN [µmol/l]	DIP [µmol/l]	DIN/DIP	Chl.a [mg m−3]	DIN [µmol/l]	DIP [µmol/l]	DIN/DIP	Chl.a [mg m−3]	DIN [µmol/l]	DIP [µmol/l]	DIN/DIP	Chl.a [mg m−3]	DIN [µmol/l]	DIP [µmol/l]	DIN/DIP	Chl.a [mg m−3]	
Wismar Bay	1.07	0.10	11	1.31	0.79	0.23	3	2.18	0.71	0.28	3	4.68	1.12	0.54	2	4.68	2.11	0.57	4	5.86	
Salzhaff	2.25	0.09	26	2.63	1.30	0.58	2	2.30	2.04	1.09	2	5.00	15.69	1.25	13	14.71	8.49	0.57	15	10.42	
Warnow Estuary	9.41	0.18	53	9.17	3.52	0.47	7	14.01	7.06	0.65	11	13.18	12.66	0.91	14	7.50	10.24	1.14	9	7.11	
Barther Bodden	4.15	0.16	25	69.53	1.35	0.14	10	51.42	1.46	0.13	11	37.11	5.91	0.17	36	56.34	2.56	0.12	21	59.31	
West Ruegenscher Bodden	1.83	0.10	19	7.42	1.47	0.16	9	9.65	1.16	0.21	5	9.65	1.24	0.18	7	12.28	0.96	0.14	7	7.60	
Strelasund	2.12	0.09	23	14.74	1.81	0.14	13	17.84	0.84	0.26	3	16.48	0.77	0.26	3	24.40	1.78	0.46	4	16.78	
Greifswalder Bodden	4.79	0.09	55	14.00	2.23	0.14	16	14.80	1.39	0.24	6	16.71	1.19	0.70	2	23.59	1.75	0.50	4	20.37	
Kleiner Jasmunder Bodden	2.07	0.34	6	111.44	4.13	0.31	13	90.81	1.85	0.18	10	67.27	2.00	0.15	13	70.08	4.41	0.31	14	83.21	
Nord Ruegenscher Bodden	1.52	0.09	16	18.44	1.17	0.14	9	19.77	1.05	0.11	9	21.56	1.47	0.21	7	24.62	1.27	0.25	5	19.45	
C 	Fall	
	October	November	
	DIN [µmol/l]	DIP [µmol/l]	DIN/DIP	Chl.a [mg m−3]	DIN [µmol/l]	DIP [µmol/l]	DIN/DIP	Chl.a [mg m−3]	
Wismar Bay	1.41	0.34	4	3.68	5.39	0.55	10	3.17	
Salzhaff	8.59	0.20	44	6.13	30.67	0.19	159	4.73	
Warnow Estuary	5.51	0.75	7	2.75	35.56	1.03	34	2.86	
Barther Bodden	2.14	0.12	18	57.22	17.32	0.18	96	49.62	
West Ruegenscher Bodden	2.27	0.25	9	6.33	4.16	0.21	20	9.40	
Strelasund	3.13	0.30	10	15.36	5.81	0.32	18	19.13	
Greifswalder Bodden	4.79	0.55	9	18.77	4.91	0.52	9	15.08	
Kleiner Jasmunder Bodden	5.76	0.16	36	96.62	27.31	0.12	219	73.16	
Nord Ruegenscher Bodden	2.34	0.16	15	15.51	3.62	0.16	23	11.63	

During March and April, high nitrogen discharges initiate blooms in coastal waters; the average DIN:DIP ratio ranged from 26 to 720, indicating a P limitation (Table 2). In May, most water bodies show a shortage of DIP. In June, phytoplankton growth reliance begins to change towards a shortage of N. During July, the majority of the waterbodies show an average DIN:DIP ratio between 2–10, indicating an N shortage. In the Wismar Bay, Nord/West Rügener Bodden, and Strelasund and Greifswalder Bodden, this situation remained the same through August and September (Table 2), whereas the Salzhaff and Warnow Estuary returned to an N/P ratio of close to 16:1 during this time. Only the shallow Kleiner Jasmunder Bodden and the Barther Bodden were exceptions. Here, the Chl. a concentrations decreased from May until July and absolute dissolved inorganic nutrient concentrations fluctuated but were relatively high (Table 2). Results indicate that phytoplankton growth is not controlled by the availability of nutrients but rather by other parameters, such as turbidity and light.

The GWB is representative of coastal waters due to its short availability of P in the spring and a shortage of N between July and September.

This is shown for GWB in Fig. 4. Between March and September, the Chl. a concentration shows a close positive correlation to TN and TP concentrations.

Figure 4 Relationships between total nutrient concentration and Chl. a concentration on a monthly basis in GWB (A–K).

This has several implications: (1) nutrients have a controlling function for phytoplankton biomass and as a consequence, measures that reduce nutrient concentrations in the water (such as mussel farming) will cause a reduction in phytoplankton concentrations and have a positive effect on water quality; (2) especially between July and September, the availability of N seems to control the productivity; summer is exactly the time when mussels show the fastest growth and are most efficient at removing nutrients from the water column via phytoplankton filtration and uptake.

Improvement of water quality during summer

Chl. a concentration and Secchi depth are related during the summer (the main grow out season for mussels) and therefore directly connected to the mitigation process of mussel farming. The linear regression shows a strong negative relationship between total nitrogen and Chl. a concentrations during July up until September (Fig. 5A). Secchi depth can be better explained by Chl. a than total suspended solid concentration (TSS) (Figs. 5B, 5D). Autocorrelation between TSS concentration and phytoplankton biomass results in a close regression coefficient because biomass is a natural part of TSS. The stronger coefficient of determination, however, affirms that Secchi depth can be better described by Chl. a concentration. Based on the prior linear regression model between TN and Chl. a concentration, a regression model describing the relationship between TN and Secchi depth was established to describe water quality improvements when nitrogen is mitigated (Fig. 5C).

Figure 5 Linear relationships between Chl. a, total nitrogen concentration, total suspended solids and Secchi depth.

(A) Chl. a concentration (Y) as a function of total nitrogen concentration (X). (B) Secchi depth as a function of Chl. a concentration. (C) Secchi depth as a function of total nitrogen concentration. (D) Secchi depth as a function of total suspended solid concentration. Functions are found by single linear regression on logarithmically transformed data (ln(Y ) = a + b × ln(X) for the summer months (July–September, 2010–2018) in Greifswald Bay.

According to linear regression model A (Fig. 5), Chl. a concentration can be reduced to approx. 1.4 mg m−3 if total nitrogen concentration is reduced to the threshold value of 16.9 µmol l−1. Based on the relationship between Secchi depth and TN concentration (C), Secchi depth would increase by approx. 70 cm. Given that GWB has a monthly water exchange rate, the reduction effort has to be accomplished on a monthly basis.

Monthly nitrogen reductions would result in annual values of 41.6 µmol l−1 TN and 10.6 mg m−3 Chl. a concentration, accounting for a reduction of 15.5% TN and 29.8% Chl. a of the mean annual concentration.

Water quality improvement by mussel farming

In situ investigations of clearance rates (CR) of blue mussels from the two pilot farms showed overall low clearance rates compared to laboratory CR investigations. Mytilus spp. from the pilot farm 2 showed slightly higher CRs at a lower ambient Chl. a concentration (15 mg m−3) than Mytilus spp. from pilot farm 1 with a higher ambient Chl. a concentration (22 mg m−3). Laboratory tests revealed a significant decrease in CR with increasing Chl. a concentration exceeding 20 mg m−3 (Fig. 6A). Based on these results, Chl. a is an important driver effecting blue mussel CRs.

Figure 6 Clearance rates and nutrient content of cultivated blue mussels.

(A) Clearance rates of Mytilus spp. within the pilot farms (pilot farm 1 = dark green; pilot farm 2 = light green) and under laboratory conditions (white) at different Chl. a concentrations. (B) Total nutrient content in percent of blue mussels.

Nutrient contents of the mussels (wet weight total) did not differ significantly between both pilot farms, indicating that neither salinity nor Chl. a concentration influence the nutrient content (Fig. 6B).

Rapid growth rates in the first few months lead to high nutrient extraction capacities per month. The first settling of mussel larvae at pilot farm 1 was recorded in July. During August, juvenile mussels had an average total wet weight of 17.78 ± 5.44 mg per mussel, while in September the average wet weight already reached 147.01 ± 28.40 mg per mussel. During August and September, TN concentrations in GWB ranged between 41.5 to 40.0 µmol l−1 (equaling approx. 1,700 t N within GWB).

Subsequently, if 50% of the GWB was covered with mussel farms, it would take up 257 t N and reduce TN concentration to 34.6 µmol l−1 by September (Table 3). According to the linear regression model (Table 3C), reduced nitrogen would increase the Secchi depth by 7.83 cm. Our results also show that if mussel farming is used locally in areas with a low water exchange rate, the effect of mussel farming can be extended. Based on the other water bodies that have no direct water exchange with the Baltic Sea (Table 1), we assumed that the Hagnesche Wiek (location of pilot farm 1 (Fig. 1)) has a lower water exchange rate of approx. 6–7 a−1. Here, the results show that the TN concentration could be reduced to 32.1 µmol l−1 and Secchi depth would increase by 10.6 cm (Table 3).

In addition to water transparency improvements seen by reducing nitrogen concentration, the mussels improve water transparency by removing phytoplankton. Taylor et al. (2020, in submission) calculated an average depletion of 5% from autumn 2017 to summer 2018 for pilot farm 1. According to the linear regression model (Fig. 5B), the reduction of Chl. a concentration from a mussel farm occupying 50% would increase Secchi depth by an additional 13.3 cm, resulting in a total improvement of ∼20 cm.

Our results clearly show that the cultivation of small sized mussels in low salinity environments must cover a substantial amount of surface area to have a severe impact on water quality. This, however, can create a tradeoff between potential improvements in water quality and negative effects emerging from the mussel farm.

Table 3 Reduction of TN concentration by mussel farming and associated water transparencyimprovement in summer (Aug/Sept).

For the Hagensche Wiek, monthly reduction of nitrogen and increase in Secchi depth wereadded together, because of its water exchange rate of 6–7 a−1.

Water body	Farm size [% of the water body surface]	Nitrogen retention by mussel growth [t]	Reduction of TN concentration [µmol l−1]	Secchi depth improvement by nitrogen retention [cm]	
		Aug.	Sept.	Aug.	Sept.	Aug.	Sept.	
GWB	1	0.5	5.1	40.7	40.6	0.64	0.50	
10	5.1	51.4	40.6	39.5	0.71	1.18	
25	12.9	128.5	40.5	37.7	0.85	3.51	
50	25.7	257.0	40.2	34.6	1.13	7.83	
Hagensche Wiek	1	0.0	0.1	39.8	0.8	
10	0.1	1.3	38.4	1.66	
25	0.3	3.1	36.1	5.9	
50	0.6	6.3	32.1	10.6	

Potential negative ecological effects of mussel cultivation in GWB

The primary negative effects of suspended mussel cultivation are nutrient enrichment of benthic habitats and the spread of pathogens.

While sediment TOC was not significantly influenced by the mussel farm in GWB (Fig. 7), during 2018 an increasing trend of TOC was indicated beneath the pilot farm. This might be explained by the fact that during 2018, the mussel farm was fully stocked (resulting in a higher bio deposition rate). Additionally, by the end of the summer, 80% of the mussel stock was lost after mussels detached and sunk, inadvertently increasing overall organic content at the seafloor.

Bio deposition was higher over the summer at pilot farm 1 than at the reference site. This was also confirmed by results from the Kiel Farm (Fig. 7). Results obtained from the commercial farm, however, showed lower bio deposition compared to GWB, even though the commercial farm was bigger in size and stock density. It has to be mentioned that measurements were not obtained at the same time. While GWB measurements were taken in June, measurements at the commercial farm were taken in October. Therefore, it can be assumed that in GWB the sedimentation of phytoplankton adds to the content of bio deposition produced by mussels, leading to significant increased values.

Figure 7 TOC of surface sediment and bio deposition rates within the pilot farm 1.

(A) Surface sediment TOC at the pilot farm 1 (green) compared to reference measurements (white) from 2017 to 2019 divided by seasons and presence of mussel stock. (B) TOC sedimentation rate in the water column in pilot farm 1 (green) and at a reference site (white). For comparison, results from the commercial mussel farm in Kiel are added.

The Vibrio spp. concentration in cultivated blue mussels was 1.07 104 ±1.75 103 CFU g−1 (Fig. 8). Higher concentrations of Vibrio spp. were found in the surface water and sediment at the reference points. However, sediment trap samples showed increased Vibrio spp. concentrations in the pilot farm 1 compared (Fig. 8).

Figure 8 Vibrio spp. appearance within the pilot farm 1.

(A) Vibrio spp. concentration (CFU 100 ml−1) at the surface water at the pilot farm 1 compared to a reference measurement. (B) Vibrio spp. concentration (CFU 100 ml−1) in the water column. Samples were taken via a sediment trap located in the pilot farm and at a reference point. Labeled in red presents the Vibrio spp. concentration (CFU g−1) detected in the cultivated blue mussels. (C) Vibrio spp. concentration (CFU 100 g−1) measured in the surface sediments beneath the pilot farm compared to reference measurement.

The results show that Vibrio bacteria are concentrated in feces and pseudo feces in the water column within the mussel farm, and concentrations in sediments are lower. This may be an indication that bio deposits are further distributed and not concentrated directly beneath the mussel farm.

Economic assessment of mussel cultivation in GWB

Based on the expected mussel yields in GWB, our results show that mussel mitigation with the revenues from feed mussel sales and nutrient removal will not be profitable in GWB (Fig. 9). Even though annual costs decreased with increasing production volume, the net balance remained negative. Revenues from nutrient reduction can only narrow the gap towards cost neutrality. In GWB, cost neutrality is not reached because the small sized mussels can only be sold as feed mussels for a minor sales price of 0.06 € kg−1. In comparison, fresh blue mussels for human consumption could be sold for 11.0 € kg−1. According to Buer et al. (2020a), mussel cultivation for human consumption is only possible at a salinity threshold of 12 PSU, indicating that mussel mitigation in Wismar Bay and Salzhaff and Warnow Estuary could be profitable if mussels can be sold for human consumption instead of animal feed.

Figure 9 Annual sales cost assessment for mussel cultivation in GWB and its comparision with reduction costs of land based measures.

(A) Total annual costs per ha mussel farm (red line) and annual sale cost balance for two scenarios (black and black dotted line). Scenario I (black dotted) represents the annual sale cost balance if the mussel farm would generate income by selling feed mussels only. Scenario II represents the annual sales cost balance if the mussel farm generates income by feed mussel sales and nutrient removal derived by removal costs of a WWTP. (B) Nitrogen (red line) and phosphorus (blue line) reduction cost (€ kg−1) per farm size compared to other mitigation measures (dotted box) derived from Gren, Jonzon & Lindqvist (2008).

In GWB, a farm size of approx. 200 ha would reduce the cost gap to under 1,000 €. This gap could be closed if water transparency improvements were compensated, as well.

Depending on the production volume, nutrient removal costs would range between 40 to 814 € kg N−1 and 899 to 18,000 € kg P−1. Even if mussel production volume increases, nitrogen extraction by mussels is more expensive than other land-based measures (Fig. 9B). Mussel mitigation becomes logical when nutrient pollution is caused by internal processes or when land-based measures have reached their feasibility limit.

Discussion

During recent years, suspended mussel farming has progressed as a novel approach to mitigate eutrophication and support underwater vegetation (Lindahl & Kollberg, 2008; Schernewski, Stybel & Neumann, 2012; Friedland et al., 2019a; Petersen, Loo & Taylor, 2019; Timmermann et al., 2019).

So far, reductions in nutrient inputs through the EU’s Urban Waste Water Treatment Directive, the modernization of WWTP, and changing agriculture practice (Munkes, 2005; Selig et al., 2006; Boesch, 2019) have not been enough to improve the ecological status in regards to Chl. a concentration and water transparency. Instead, after 2000 there was a stagnation in nutrient reductions across most inner coastal waters of north east Germany and WFD targets were not met.

Nevertheless, most water bodies could potentially reach the target values demanded by the WFD if nutrient loads were further reduced and internal restoration measures implemented. This is facilitated by the fact that climate change in north east Germany will most likely reduce diffuse nutrient run off and nitrogen pollution further. Longer droughts and less rain in the summer might limit crop production and the use of nitrogen fertilizer (Spekat, Enke & Kreienkamp, 2007; Jacob et al., 2008; Borken & Matzner, 2009). This could further shorten the gap to the desired annual target values and make internal mitigation more effective (Williges et al., 2017). Overall, the results show that in the summer, the availability of nitrogen controls phytoplankton productivity. Therefore, by further shortening the nitrogen supply during the summer, phytoplankton growth can be hampered and water transparency can be improved (Nielsen et al., 2002).

Mussel farming as an internal mitigation measure is capable of reducing nutrients up to 0.6–3.01 t N ha−1 and 0.03–0.17 t P ha−1 in Danish waters, depending on the farming strategy and harvesting time (Petersen et al., 2014; Taylor et al., 2019). Timmermann et al. (2019) showed that blue mussel farming in the eutrophied Danish Skive fjord decreased summer Chl. a concentration by 30% and light attenuation up to 14%. Improvements were also obtained in Sweden and Germany (Lindahl et al., 2005; Schröder et al., 2014).

However, our results have shown that mussel cultivation in shallow coastal waters can be subject to risks and success depends very much on the ecosystem and prevailing conditions.

The osmotic stress based on the salinity gradient in the Baltic has the most dominant effect on blue mussel growth and health (Buer et al., 2020a; Buer et al., 2020b; Maar et al., 2015; Riisgård, Bøttiger & Pleissner, 2012; Westerbom, Kilpi & Mustonen, 2002) and therefore also on the mitigation effect.

Another factor which can affect the mitigation potential is the loss of mussel stock. Loss of mussels can be caused by different events e.g., by predators such as eider ducks, damage or loss of farm structures as well as changing environmental parameters such as high water temperatures during summer (Varennes et al., 2013; Jones et al., 2010; Ross & Furnes 2000;). Especially, in shallow coastal waters with limited water exchange rates high water temperatures can appear during summer. Fly and Hilbish (2013) stated that the energetic balance of M. trossulus is affected by water temperatures above 17 °C. Other studies have shown that byssus threads are up to 60% weaker at water temperatures of 25 °C than at 10 °C (Newcomb et al., 2019). Westerbom et al. (2019) concluded that temperature stress coupled with continuous salinity stress may also cause increased mortality. Therefore, there is an increased risk at low salinities that mussels detach, die off and increase the organic pressure on the sediment below, while at the same time decrease mussel harvest.

In addition, enclosed shallow coastal waters often suffer from internal loading and resuspension, fuelling high phytoplankton biomass and particular matter (Schiewer, 1997; Selig et al., 2006) which can also impact the effectiveness of mussel mitigation (Schulte, 1975; Riisgard, 1991).

Mesocosm and laboratory experiments showed lower CRs at high Chl. a concentrations (Fig. 6A), supporting previous studies (Clausen & Riisgård, 1996; Riisgård, Kittner & Seerup, 2003; Soon, Denil & Ransangan, 2016). Clearance rates in mesocosm experiments were significantly lower than under laboratory conditions, most likely because mussels under laboratory conditions were fed a Rhodomonas sp. diet optimal for Mytilus spp. (Fernández-Reiriz et al. 2015). Three of the assessed water bodies (Barther Bodden, Kleiner Jasmunder Bodden, Nord Rügener Bodden) exceed Chl. a concentration of 20.0 mg m−3 throughout the entire summer, while others, e.g., Greifswalder Bodden andf Strealsund, exceed these only at the end of summer (August, September).

Comparing results of the present study with available models, e.g., the Farm Aquaculture Management System (FARM), it becomes apparent that the effectiveness of mussel mitigation is hampered by Chl. a concentrations above 20 mg m−3. For example, the FARM model calculates an Assessment of Estuarine Trophic Status (ASSETS) score of “moderate” in regard of the late summer Chl. a concentrations of GWB (>20.0 mg m−3). If the Chl. a concentration is set to 10.0 mg m−3 the calculate ASSETS score reaches “good”.

Even though there was no significant difference in nutrient content (total DW was detected between blue mussel samples from both pilot farms), mussel samples from pilot farm 1 had a lower nutrient content in total WW. Mussels of this farm had a higher water content due to lower salinity, supporting the results of Maar et al. (2015). With increasing salinity, the nutrient storage capacity of blue mussels increases, improving the mitigation effect.

One of the greatest concerns regarding intense mussel farming is enhanced local bio deposition beneath mussel farms (Nizzoli et al., 2005; Giles, Pilditch & Bell, 2006; Callier et al., 2009; Giles et al., 2009). Surveys of benthic conditions beneath mussel cultivation sites in shallow coastal embayment’s (2 m–10 m water depth) in Canada revealed that after a production increase of 43% (approx. 20 ×103 t y−1 in 2001) organic matter (TOC) also increased significantly. Furthermore, after a cultivation period of two years, approx. 50% of the sediment beneath the cultivation site was permanently anoxic and dominated by white sulfur bacteria mats (Cranford et al., 2009; Hargrave, 2015). Due to the small size and fluctuating stock density of the pilot farms in this study, there was no significant enrichment of sediment TOC.

Increased bio deposition in GWB during the summer and at the larger farm in the Kiel Fjord, however, support previous studies and point towards a risk of increased TOC if farm units exceed a size specific for the ecosystem.

Filter feeders like blue mussel also accumulate pathogens from the surrounding water (Canesi et al., 2002). Especially in the summer months, when water temperatures are high, Vibrio species are highly abundant in coastal waters (Eiler et al., 2007; Baker-Austin et al., 2013). Collin et al. (2012) stated that blue mussels are able to accumulate around 98% of marine Vibrio species within a 24-hour period and, due to their robust immune systems, can eliminate these pathogens (Balbi et al., 2013; Tanguy et al., 2013). However, our results show that Vibrio spp. are concentrated by mussels and released into the water column through feces and pseudo feces, indicating that the mussel immune system was not able to effectively erase Vibrio bacteria. In this study, total Vibrio spp. abundance obtained from the mussel’s tissue was approximately 2 times higher than 102–103 of CFU per gram described in Huss (1997). High concentrations paired with a reduced immune response caused by environmental stressors (Rahman et al., 2019) might explain the release of viable bacteria. Pietrak, Molloy & Bouchard (2012) also argued that it is possible that Vibrio bacteria are released in the form of pseudo feces production without passing through the digestive system. In both cases, mussels and mussel farms could act as a sink of potentially pathogenic Vibrio bacteria in the water column (Callol et al., 2015; Ina-Salwany et al., 2019). Even though high concentrations were observed in the bio deposits in the water column, the sediment concentrations did not increase compared to the reference site.

In both cases, when considering increased bio deposition and Vibrio concentration, the pilot farm was too small to have an impact on the sediment beneath. Since it can be expected that bigger farms in inner coastal waters will have a significant impact on the environment, more research regarding ecological and pathological tradeoffs is needed.

The profitability of mussel cultivation can be an important factor in the selection and acceptance of a mitigation measure. Buer et al. (2020a) described the effects of the salinity gradient along the German Baltic coast on blue mussel farming and its economic implications. The authors state that mussel farming here can be profitable only if blue mussels are sold for human consumption. The threshold that this is still feasible is above 12 PSU, beyond which point mussel growth is too slow to be profitable (Buer et al., 2020a). Beneath 12 PSU, alternative sale strategies, such as animal feed (Schernewski, Stybel & Neumann, 2012) or compensation for nutrient removal are proposed.

The results of this study show that even if mussel cultivation generates revenues from feed mussel and nutrient extraction (based on the removal costs of a WWTP), the sales cost balance remains negative and annual investment and running costs cannot be recovered. If blue mussels cannot be sold for human consumption, nutrient mitigation will always be a no-income measure, making it less attractive. Therefore, further funding schemes are necessary. Another option could be a nutrient trading system where the polluter pays for the emission. This is easy to accomplish for point source nutrient inputs, but becomes more challenging if diffuse sources are included (Ferreira & Bricker, 2016). Additionally, mussel farming could generate additional income through the improvement of water transparency and its implications on ecosystem services. Tuya, Haroun & Espino (2014) described the monetary values of seagrass meadows that act as nursery grounds for commercially interesting fish species. The results of this study revealed an average monetary value of seagrass meadows of 95.75 € ha−1 y−1. The same approach could be applied for German coastal waters and be used to compensate mussel cultivation for increasing water transparency.

Conclusion

This study showed that mussel cultivation in inner coastal waters can act as a nutrient and phytoplankton sink during the summer and, potentially increase water transparency and light availability for submerged macrophytes. However, depending on physical and biological restrictions, mussel cultivation will have different improvement efficiencies mainly due to the predominant salinity level. Estimations for the case study area GWB, which is with 6–7 PSU a marginal cultivation site, have shown that if 50% of GWB are used for mussel cultivation, 257 t TN can be bound in mussel biomass. This reduction could increase Secchi depth by 7.8 cm.

However, due to the limited mussel growth, this mitigation measure cannot be economic for this site. Further, the mitigation potential can also be affected by unpredicted mussel loss caused by high temperatures. Even though this study showed no negative environmental impacts by mussel farming, it can be assumed that shallow eutrophic coastal waters are especially vulnerable if production volumes were increased. For coastal waters with low salinity levels, a combination of different internal mitigation measures would be recommended.

A higher concentration of Vibrio spp. was detected in bio deposits but not in the sediment below the farm in GWB. Nonetheless, further research is needed to investigate if large mussel farms can act as a sink for human pathogens like Vibrio spp. This is especially important for systems that already suffer from high Vibrio spp. concentrations potentially causing a conflict for bathing tourism. Mussel cultivation reaches highest mitigation potential at salinities above 12 PSU and at Chl. a concentrations below 20 mg m−3. However, including economic estimations, mussel farming will not be a cost neutral mitigation measure based on the current market prices for feed mussels and compensation compared to nutrient removal cost of WWTP.

Supplemental Information

Supplemental Information 1 Raw data

Bio-chemical data of inner coastal waters and data which was used to determine clearance rates and nutrient content of cultivated Mytilus spp. Further raw data shows data which was used to determine TOC content beneath mussel farms, bio deposition rates and Vibrio bacteria concentration in mussel farms. Last but not least, the raw data is showing the economic assessment conducted for the mussel farm.

Click here for additional data file.

We would like to thank MV Weber (LUNG-MV) for the provision of environmental data. Additionally, we are grateful to the native English speaker M Nepf for checking the manuscript.

Additional Information and Declarations

Competing Interests

Author Contributions

Data Availability

The authors declare there are no competing interests.

Lukas Ritzenhofen conceived and designed the experiments, performed the experiments, analyzed the data, prepared figures and/or tables, authored or reviewed drafts of the paper, and approved the final draft.

Anna-Lucia Buer, Sven Dahlke and Annemarie Klemmstein conceived and designed the experiments, performed the experiments, authored or reviewed drafts of the paper, and approved the final draft.

Greta Gyraite conceived and designed the experiments, performed the experiments, analyzed the data, authored or reviewed drafts of the paper, and approved the final draft.

Gerald Schernewski conceived and designed the experiments, authored or reviewed drafts of the paper, and approved the final draft.

The following information was supplied regarding data availability:

The raw data is available in the Supplemental File.

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
