# Peer review of "­Blue mussel (Mytilus spp.) cultivation in mesohaline eutrophied inner coastal waters: mitigation potential, threats and cost effectiveness"

_PeerJ, doi:10.7717/peerj.11247_

## Round 0.1 · original submission · Major Revisions

The reviewers have raised a number of questions that need to be addressed, particularly to do with the breadth of data presented and that this tends to distract from the key findings on the impact of mussel farms. I also felt that this paper could be more carefully focused and significantly reduced in length without losing critical information. In undertaking these changes, I would also encourage you to seek some assistance from a fluent English speaking colleague - there are a large number of small grammatical changes required.

Reviewer 1 ·

Basic reporting

line 56-58. “The mitigation potenial...” I believe that it can be hard for many readers to understand this sentence. Since it is a quite important finding I suggest that you rephrase and develop the reasoning.

line 93. Did Stadmark and Conley, 2011 say that mussel cultivation is one of the most promising measures?

line 99-102. It is not clear to me what you are meaning with these sentences?

line 106-107. I do not believe that this is relevant to state what is possible. There are many places where you don´t find Dreissena and Mytilus is very small.

Line 132 It seems strange to begin this very long sentence with therefore.

Line 196 It also seems strange to begin this sentence with therefore.

Line 206 Same goes for this sentence

Experimental design

There are many different types of data, investigations and experiments that are presented in this paper. It makes it a bit hard to read and grasp. I really do not see the point with spending so much effort and space in the status environmental status of the inner coast waters. It seem rather like background information and could perhaps be moved to an Appendix?
I am also a bit skeptical a bout the economic section which has been much more thoroughly examined many times before.

However, the data from the pilot farms is important. I suggest to focus on presenting and discussing the results from these farms.

As I interpret it, the main knowledge gap you fill is that you can describe the mitigation potential for mussel farms in mesohaline inner coastal waters. Your result points that out that the overall potential is quite small. Even if scaling up our farms many magnitudes the effect will be very small. On top of that you describe risks with aggregation of Vibero, that harvest fails and weak economic potential. This can be clearer in the conclusion and abstract.

Validity of the findings

line 442-450. Subsequently, if 50% of the GWB was covered with mussel farms. , they would take up 257 t N and reduce TN concentration to 34.6 μmol l-1 within September (Tab. 3). According to the linear regression model (Tab. 3, C), reduced nitrogen would increase the Secchi depth by 7.83 cm...
and line 457-460
I consider this a main finding of the study. This should be highlighted in abstract and conclusion

line 470. Farms loosing their stocks has been very common in Baltic mussel farming trials, I believe that this result also should be highlighted and discussed more since it has huge potential impact on the mitigation function.

line 517-553 Seems to fit better as introduction rather then discussion

line 575-579 These findings should be discussed in relation to models and projection using lab values e.g. FARM

Additional comments

I believe that this is an important study. However it can be clearer and more focused.

·

Basic reporting

The paper by Ritzenhofen et al. „Blue mussel (Mytilus spp.) cultivation in mesohaline eutrophied inner coastal waters: mitigation potential, threats and cost effectiveness“ is a very timely contribution to improve the knowledge base of the novel aquaculture activities in the Baltic Sea and beyond.

The paper is very well and clearly written. I only have the following minor recommendations:

In the abstract section, please add the density of Vibrio spp. in the reference areas (sediments/mussels). If their density was zero, just spell this clearly out. From the abstract section it is not clear if you measured Vibrio in the biodeposits (e.g. using sediment trap to collect biodeposits) or you sampled bottom sediments (including reworked biodeposits and other material) or both. Importantly, in the discussion section you actually say that “Even though high concentrations were observed in the bio deposits in the water column, the sediment concentrations did not increase compared to the reference site.” This suggests that the Vibrio statement above is a bit of oversell (i.e. not ecologically relevent at larger spatial scales) and instead you should use the latter statement in the abstract.
Perhaps, you need to spell out the scenario of mitigation compensation (how the scenario was priced) as this background information very much defines whether the activity is economically feasible or not.
Add to the introduction or method section information about the prevalence/rarity of natural mussel beds in your studied ecosystems.
Line 93: Please cite also to a recent publication that dealt with the mitigation potential of Mytilus in the whole Baltic Sea region: Kotta, J.; Futter, M.; Kaasik, A.; Liversage, K.; Rätsep, M.; Barboza, F.R.; Bergström, L.; Bergström, P.; Bobsien, I.; Díaz, E.; Herkül, K.; Jonsson, P.R.; Korpinen, S.; Kraufvelin, P.; Krost, P.; Lindahl, O.; Lindegarth, M.; Lyngsgaard, M.M.; Mühl, M.; Sandman, A.N.; Orav-Kotta, H.; Orlova, M.; Skov, H.; Rissanen, J.; Šiaulys, A.; Vidakovic, A.; Virtanen, E. 2020. Cleaning up seas using blue growth initiatives: mussel farming for eutrophication control in the Baltic Sea. STOTEN, 709, 136144.
In addition, you miss another important publication in the field, which very much relates to the current paper: Andreas Michael Holbach et al. 2020 A spatial model for nutrient mitigation potential of blue mussel farms in the western Baltic Sea. https://doi.org/10.1016/j.scitotenv.2020.139624
Line 108: Do you mean a relative small negative environmental impact?
Line 164: Sediment types consist mostly of ....
Add to the discussion section a statement that says how often you expect in your study areas conditions with chl > 20. Now it is not clear if this is a short term phenomenon or some estuaries can experience such conditions troughout the summer.
In the conclusion section, a statement about the effects of Vibrio at the ecosystem level is too far reached as the elevated densities were measured in biodeposits only and not in the sediment.

Experimental design

No comments

Validity of the findings

No additional comments, see my minor comments under the section of basic reporting.

Additional comments

No comments.

---

## Round 0.2 · accepted · Accept

The care and attention paid during revision are appreciated and the reviewers are supportive of publication, but please note a couple of final reviewer comments and make any changes required (L97).

Reviewer 1 ·

Basic reporting

I believe that the paper has improved a lot. I still believe it would have been easier to read and comprehend if less focus had been on the economics and the background environmental status of the coast. However, I am happy with the changes

I do not really understand what you mean with threats in the title? Mitigation threats or farming threats or environmental threats?
Line 66. This is Vibrio sink hypothesis is interesting I hope you investigate it!
Line 97. I guess that the reference Wikström et al 2020 would rather support the previous sentence in line 91-94. I found a similar citation error in the previous version too, please check that the references are correct.

Experimental design

no comment

Validity of the findings

no comment

Additional comments

no comment